# AGREEing on Clinical Practice Guidelines for Autism Spectrum Disorders in Children: A Systematic Review and Quality Assessment

**DOI:** 10.3390/children9071050

**Published:** 2022-07-14

**Authors:** Yasser S. Amer, Shuliweeh Alenezi, Fahad A. Bashiri, Amel Hussain Alawami, Ayman Shawqi Alhazmi, Somayyah A. Aladamawi, Faisal Alnemary, Yasser Alqahtani, Maysaa W. Buraik, Saleh S. AlSuwailem, Shahad M. Akhalifah, Marcela Augusta de Souza Pinhel, Melanie Penner, Ahmed M. Elmalky

**Affiliations:** 1Pediatrics Department, King Khalid University Hospital, King Saud University Medical City, Riyadh 11451, Saudi Arabia; 2Clinical Practice Guidelines and Quality Research Unit, Quality Management Department, King Saud University Medical City, Riyadh 11451, Saudi Arabia; 3Research Chair for Evidence-Based Health Care and Knowledge Translation, Deanship of Scientific Research, King Saud University, Riyadh 11451, Saudi Arabia; 4Alexandria Center for Evidence-Based Clinical Practice Guidelines, Alexandria University, Alexandria 5424041, Egypt; 5Guidelines International Network, Perth PH16 5BU, Scotland, UK; 6Department of Psychiatry, College of Medicine, King Saud University, Riyadh 11451, Saudi Arabia; 7Pediatric Neurology Division, Pediatrics Department, College of Medicine, King Saud University, Riyadh 11451, Saudi Arabia; fbashiri@ksu.edu.sa; 8Johns Hopkins Aramco Healthcare, Dhahran 34465, Saudi Arabia; awamiah95@gmail.com (A.H.A.); maissaa.buraik@jhah.com (M.W.B.); 9Developmental Pediatric Department, Children’s Hospital, King Saud Medical City, Ministry of Health, Riyadh 12746, Saudi Arabia; dr.ayman11@gmail.com; 10King Abdullah bin Abdulaziz University Hospital, Riyadh 11564, Saudi Arabia; saladmawi@kaauh.edu.sa; 11College of Medicine, Princess Nourah Bint Abdulrahman University, Riyadh 11564, Saudi Arabia; 12Autism Center of Excellence, Riyadh 11564, Saudi Arabia; falnemary@acesaudi.org (F.A.); ssalsuwailem@gmail.com (S.S.A.); shahad.alkh@gmail.com (S.M.A.); 13King Fahad Armed Forces Hospital, Jeddah 23311, Saudi Arabia; yasser.alqahtani92@hotmail.com; 14Department of Health Science, Ribeirao Preto Medical School, University of São Paulo, Ribeirao Preto 14049-900, Brazil; marcelapinhel@yahoo.com.br; 15Department of Molecular Biology, São José do Rio Preto Medical School, São José do Rio Preto 15090-000, Brazil; 16Holland Bloorview Kids Rehabilitation Hospital, Toronto, ON M4G 1R8, Canada; mpenner@hollandbloorview.ca; 17Department of Pediatrics, University of Toronto, Toronto, ON M5G 1X8, Canada; 18Morbidity and Mortality Unit, King Saud University Medical City, King Saud University, Riyadh 11451, Saudi Arabia; dr_cphq_kkuh@ymail.com; 19Public Health and Community Medicine Department, Theodor Bilharz Research Institute (TBRI), Academy of Scientific Research, Cairo 3863130, Egypt

**Keywords:** autism spectrum disorder, psychiatry, pediatrics, clinical practice guidelines, systematic review, AGREE II instrument, quality assessment

## Abstract

Background: Autism spectrum disorder (ASD) is a multifaceted neurodevelopmental disorder requiring multimodal intervention and an army of multidisciplinary teams for a proper rehabilitation plan. Accordingly, multiple practice guidelines have been published for different disciplines. However, systematic evidence to detect and intervene must be updated regularly. Our main objective is to compare and summarize the recommendations made in the clinical practice guidelines (CPGs) for ASD in children released from November 2015 to March 2022. Methods: CPGs were subjected to a systematic review. We developed the inclusion and exclusion criteria and health-related questions, then searched and screened for CPGs utilizing bibliographic and CPG databases. Each of the CPGs used in the study were critically evaluated using the Appraisal of Guidelines for REsearch and Evaluation II (AGREE II) instrument. In a realistic comparison table, we summarized the recommendations. Results: Four eligible CPGs were appraised: Australian Autism CRC (ACRC); Ministry of Health New Zealand (NZ); National Institute for Health and Care Excellence (NICE); and Scottish Intercollegiate Guidelines Network, Healthcare Improvement Scotland (SIGN-HIS). The overall assessments of all four CPGs scored greater than 80%; these findings were consistent with the high scores in the six domains of AGREE II, including: (1) scope and purpose, (2) stakeholder involvement, (3) rigor of development, (4) clarity of presentation, (5) applicability, and (6) editorial independence domains. Domain (3) scored 84%, 93%, 86%, and 85%; domain (5) 92%, 89%, 54%, and 85%; and domain (6) 92%, 96%, 88%, and 92% for ACRC, NICE, NZ, and SIGN-HIS, respectively. Overall, there were no serious conflicts between the clinical recommendations of the four CPGs, but some were more comprehensive and elaborative than others. Conclusions: All four assessed evidence-based CPGs demonstrated high methodological quality and relevance for use in practice.

## 1. Introduction

Epidemiology of Autism Spectrum Disorder (ASD)

Autism spectrum disorder (ASD) is a neurodevelopmental disorder associated with difficulties in social interaction, communication, and repetitive behaviors [1]. However, individuals with ASD have a range of cognitive abilities, from substantial intellectual disability to above-average cognitive functioning, and different language skills range from non-verbal to fluent. Additionally, affected individuals may exhibit co-occurring medical diseases such as epilepsy, gastrointestinal problems, and psychiatric comorbidities, resulting in multiple layers of clinical complexity [2,3].

The American Psychiatric Association recognized autism for the first time as a distinct clinical diagnostic in the third edition of the *Diagnostic and Statistical Manual of Mental Disorders* [4,5]. Revisions to the diagnostic criteria were made to the fourth edition of the DSM in 1994, and five subtypes of autism were added as a result: autistic disorder, Asperger disorder, pervasive developmental disorder-not otherwise specified (PDD-NOS), childhood disintegrative disorder, and Rett’s disorder [5]. However, the last two conditions fall under the more general category of pervasive developmental disorders, while the first three categories make up autism spectrum disorder (ASD) [2]. Furthermore, ASD was reclassified in the fifth version of the DSM [1] in 2013 as a single condition with three severity categories termed specifiers.

Despite numerous studies, the prevalence of ASD is still not well defined, with multiple studies worldwide reporting differing rates in various continents and countries. However, the World Health Organization [6] estimated in 2019 the global prevalence of ASD to be 6.25 cases per 1000. The most recent version (with 2018 data) published by the Center for Disease Control and Prevention Agency (CDC), USA, which operates the Autism and Developmental Disabilities Monitoring Network, and assesses 11 North American states indicated that, among 8-year-old children, the prevalence of ASD ranged from 16.5 per 1000 in Missouri to 38.9 per 1000 in California. These data estimated an ASD prevalence of 23.0/1000, which is equivalent to 1 for every 44 children aged 8 years. It also showed that 80% of confirmed ASD cases were boys (4.2 times) [7].

The difference between genders also seems to be another relevant point in different populations [8,9]. It appears that boys are more affected (with a ratio of four times more frequent) when compared to girls [10]. Recent research has indicated that this explanation might be related to the under diagnosis of females with ASD, particularly when they are young [11]. In this context, the literature shows that studies have reported variable male-to-female ratios, varying from 2:1 to 5:1, documented in the literature [12,13].

In an attempt to improve this scenario and determine the real prevalence of ASD in different populations, early detection and correct diagnosis of individuals with ASD is critical. Two studies made an important discussion of this topic, pointing out that many ASD symptoms are noticeable in early childhood and allow diagnosis before 3 years of age [14,15]. However, despite this possibility of early diagnosis, there is a significant difference between the potential age of diagnosis and the actual age of diagnosis, which is often only performed when the child reaches school age [14,15].

Clinical practice guidelines (CPGs) offer guidance for enhancing patient care based on a systematic review (SR) of the available data and an evaluation of the advantages and disadvantages of different care options [16]. Many international evidence-based clinical practice guideline (CPG) programs have highlighted ASD as one of the top health priorities. The high volume of published ASD CPGs of variable quality has been quite confusing for ASD healthcare professionals and providers. The need has arisen for an updated systematic review and critical appraisal of these CPGs [17].

The AGREE II (Appraisal of Guidelines for REsearch and Evaluation) instrument is the gold standard for assessing the quality of CPGs. AGREE II has been validated, widely cited, and endorsed by a number of healthcare organizations [18,19,20]. AGREE II specifies components that CPGs should address in order to increase their quality and trustworthiness, as well as to achieve positive patient outcomes [18,20,21,22].

We devoted this study to reporting the outcomes of this SR and critically evaluating the recently released CPGs for children with ASD using the Appraisal of Guidelines for REsearch and Evaluation (AGREE) II, as it is a core step in the CPG adaptation process [21,22,23,24].

The main objective of this study was to update the SR of ASD CPGs that was published by Penner et al. that focused exclusively on diagnosis and to expand on the recommendations for treatment as well [25]. Reporting of the full CPG adaptation process with all of its steps and final results including the adapted CPG is out of the scope of this paper and will be published separately.

## 2. Materials and Methods

We started by registering the protocol for this study at the Center for Open Science (OSF) [24]. Our CPG adaptation group included two child and adolescent psychiatry consultants, a pediatric neurology consultant, three developmental behavioral pediatric consultants, a board-certified behavior analyst, a clinical psychology consultant, a senior speech and language therapist, a senior occupational therapist, and a general pediatrician and CPG methodologist. To contribute to this SR study, two international researchers in ASD joined as international collaborators.

In Saudi Arabia, there are no national CPGs for comprehensive recommendations for managing children with ASD based on evidence-based research. The Saudi Health Council, National Center for Developmental and Behavioral Disorders, and National Center for Evidence-Based Health Practice launched a national CPG adaptation project for the management of children with ASD in 2021 to address this gap and offer direction and recommendations to all pertinent stakeholders who care for these children in Saudi Arabia. The ‘KSU-Modified-ADAPTE’ was used to guide the project, which is a formal CPG adaption process including three phases: setup, adaptation, and finalization [21,22,23,24,26].

This systematic review followed the methodological guide for systematic reviews of guidelines in addition to the 2020 updated Preferred Reporting Items for Systematic Reviews and Meta-Analyses (PRISMA) statement [23,27,28]. Systematic review registration: The research team registered the protocol in the Center for Open Science (OSF) in 2021 https://osf.io/n39xz/.

### 2.1. Data Sources and Search Strategy

We systematically searched bibliographic databases, CPG databases and repositories, and databases of ASD-related national and international professional societies (Appendix A).

The search was limited to CPGs published from November 2015 until March 2022. We decided on this period of time to update the search conducted in the previous SR of CPGs, as the last systematic review was conducted in 2018 [25]. The search was performed by a CPG methodologist (YSA). To facilitate the CPG eligibility identification procedure, we used the PIPOH (Population, Intervention, Professions, Outcomes, and Health care system) model and the PICAR statement (Population, Interventions, Comparators, Attributes of the CPG, and Recommendation characteristics) (Appendix A) [21,22,23]. Titles and abstracts of the retrieved CPGs and papers matching the inclusion criteria were independently evaluated by two reviewers (A.H.A. and S.M.A.). Disagreements were resolved through focus group discussions following the retrieval and examination of full-text articles or complete CPG documents.

### 2.2. Eligibility Criteria

The following were the eligibility criteria: (1) evidence-based with described CPG development methods; (2) English or Arabic language (based on language abilities of the research team); (3) original source CPGs (de novo developed); (4) national or international scope; (5) published after October 2015; and (6) published with group or organization authorship that may be accessed from a CPG database or peer-reviewed journal. Each source CPG was only evaluated using the most recent version.

CPGs that were published in October 2015 or earlier, were not in English or Arabic, were derived from other CPGs, were presented as consensus or expert-based statements, had only one author, or had any of the aforementioned characteristics were all excluded.

### 2.3. The Capacity Building for Using the AGREE II Instrument

Through a hands-on training on evidence-based CPG standards and the use of the AGREE II instrument, the CPG methodologist delivered a workshop for capacity building for the CPG assigned appraisers. Following that, each reviewer graded their assigned CPGs. Each CPG was critically evaluated by all four reviewers. The whole CPG materials were read by all appraisers, including any revisions with any relevant supplemental information or links to internet web pages relating to CPG methodologies or CPG implementation tools. AGREE appraisers were told to document their reasoning for their scores in the “Comment” box for each item [29].

### 2.4. Quality Assessment of ASD CPGs Using AGREE II

Scope and purpose, stakeholder involvement, rigor of development, clarity of presentation, applicability, and editorial independence are among the six domains that make up the AGREE II instrument’s 23 elements [18,19,29]. The AGREE II evaluation was completed using its online version, “My AGREE PLUS”, which creates a CPG appraisal group for each CPG and translates the item ratings into domain ratings and comments. Each item was graded on a 7-point Likert scale [18,19,29]. The four AGREE II appraisers included a consultant developmental behavioral pediatrician (SAA), a consultant clinical psychologist (MWB), a senior speech and language therapist (SA), and as well as a registered pediatrician with expertise in CPG methodologies (YSA).

Group discussions were used to settle significant disparities in the AGREE II appraisers’ scores of items or questions (i.e., a difference of 3 or more). The online My AGREE PLUS system automatically calculated the standardized AGREE II domain scores or ratings. For each domain score, we agreed on a cutoff point of 70%. Following the appraisal, greater emphasis was placed on the scores of domains 3 and 5 in order to facilitate the filtration and final evaluation of the reporting quality of the included CPGs. Identical cutoff values have been reported [30,31]. The evidence bases of the included CPGs, specifically their references sections, were examined for SRs or meta-analyses, particularly Cochrane SRs, through the references section of the included source CPGs in addition to the six AGREE II domains.

### 2.5. Analysis Plan

Using the AGREE II instrument manual, we calculated scores (0–100%) for each AGREE II domain. In a comparative tabular format, the key recommendations of the eligible CPGs were summarized. According to the rating of domain 3 (rigor of development), the quality of CPGs was categorized, with a high-quality CPG earning a standardized domain rating of more than or equal to 70%, a moderate quality CPG (40–69%), and a low-quality CPG (40%).

#### Inter-Rater Analysis

To measure the amount of agreement between raters, we performed inter-rater reliability tests (IRR). To examine the amount of agreement among the four raters, we employed a percent agreement inter-rater reliability assessment test for each question in each area in the four CPGs as well as the first overall assessment’s percent agreement. In order to assess the consistency of ratings or capacity for datasets collected as clusters or sorted into clusters, we also used intra-class correlation (the second overall assessment).

One of the most used IRR techniques is intra-class correlation (ICC). We resort to this technique when we have more than two raters. A high intra-class correlation coefficient (kappa) around one indicated great similarity between standards from the same set. A low Kappa value close to 0 suggested that standards from the same set are not identical. Because we had inconsistent raters/rates, we utilized ANOVA “One-Way Random” on SPSS Statistics, version 21. We chose ICC because of the variety of numerical data from groups or clusters. This aided us in determining repeatability as well as how closely peers resemble one another in terms of particular qualities or attributes. We looked at how well two ordinal scale categories agreed. Because the data came from an ordered scale, we employed Weighted Kappa (Quadratic Weights). The weights are determined as follows. Using the Cohen’s Kappa notation, we selected linear weights since the difference between the first and second categories was equivalent to the difference between the second and third categories, and so on. The Kappa (K) statistic [32,33] is used to quantify agreement: K equals 1 when there is complete agreement between the categorization systems, K equals 0 when there is no agreement greater than chance, and K is negative when agreement is worse than chance. Appendix A shows how the K value might be interpreted [34].

## 3. Results

A total of 165 records were retrieved, with 161 being excluded based on eligibility criteria. According to the PRISMA flowchart [27], only four original source CPGs were found to be eligible for the quality assessment step (Appendix A). Two reviewers (A.H.A., S.M.A.) conducted the screening, and two additional reviewers (S.M.A., Y.S.A.) discussed any discrepancies.

### 3.1. Characteristics of Included ASD CPGs

Table 1 outlines the characteristics of the four eligible CPGs. The CPG developer organizations included the Australian Autism CRC (ACRC), the Ministry of Health New Zealand (NZ), the National Institute for Health and Care Excellence (NICE), and the Scottish Intercollegiate Guidelines Network, Healthcare Improvement Scotland (SIGN-HIS). All these organizations were from high-income countries.

### 3.2. Quality Assessment of the ASD CPGs

Table 2 and Figure 1 and Figure 2 summarize the AGREE II standardized domain ratings.

#### 3.2.1. Domain 1: Scope and Purpose

Domain 1 scores ranged from 93 to 99%. All four CPGs were higher than 70% (ACRC = 99%, NICE = 96%, NZ = 93%, and SIGN-HIS = 97%) [35,36,37,38].

All of the four appraised CPGs have specifically reported their overall objective or purpose, health questions, and patient population clearly [35,36,37,38].

#### 3.2.2. Domain 2: Stakeholder Involvement

Domain 2 scores ranged from 71 to 99%. All four CPGs were higher than 70% (ACRC = 99%, NICE = 71%, NZ = 71%, and SIGN-HIS = 79%) [35,36,37,38].

All of the four appraised CPGs have specifically reported their CPG development groups, target end users, and patient views and preferences considerations [35,36,37,38].

#### 3.2.3. Domain 3: Rigor of Development

AGREE II scores for Domain 3 were from 84 to 93%. In domain 3, the scores of all four CPGs were more than 70% (ACRC = 84%, NICE = 93%, NZ = 86%, and SIGN-HIS = 85%) [35,36,37,38].

All of the four appraised CPGs reported using systematic methods to search for evidence, evidence inclusion and exclusion criteria, evidence strengths and limitations, methods for formulating the recommendations, considering the risks and benefits in formulating the recommendations, links between the recommendations and supporting evidence, external review, and a clear process for review and update [35,36,37,38]. Both NICE and SIGN-HIS CPGs documented using the Grading of Recommendations, Assessment, Development, and Evaluations (GRADE) method in their formal development methodology [37,38].

#### 3.2.4. Domain 4: Clarity of Presentation

The AGREE II scores for domain 4 ranged from 93 to 97%. The scores of all four CPGs were above 70% (ACRC = 93%, NICE = 97%, NZ = 94%, and SIGN-HIS = 93%) [33,34,35,36]. All of the four appraised CPGs reported specific and clear detailed recommendations, different management options, and key or summary recommendations [35,36,37,38].

Appendix A provides a comparison between the recommendations of the four assessed CPGs in a recommendation matrix format.

#### 3.2.5. Domain 5: Applicability

The AGREE II scores for domain 5 reported from 54 to 92%. The scores of three CPGs were higher than 70% (ACRC = 92%, NICE = 89%, and SIGN-HIS = 85%) [30,37,38]. All the four appraised CPGs reported noted implementation considerations including facilitators and barriers to application, different CPG implementation tools, resource implications, and auditing criteria [35,36,37,38].

#### 3.2.6. Domain 6: Editorial Independence

Scores for domain 6 ranged from 69 to 92%. The scores of three CPGs were more than 70% (ACRC = 92%, NICE = 92%, and SIGN-HIS = 92%) [30,32,33]. All appraised CPGs reported the funding body and the competing interests of the CPG development group [35,36,37,38].

#### 3.2.7. Overall Assessment

The AGREE II standardized domain scores ranged from 88 to 96% for the first overall assessment. Scores of higher than 70% were received by all four CPGs, which correlated to excellent performance across all six AGREE II domains.

#### 3.2.8. Recommending the ASD CPGs for Use in Practice

The second overall assessment (i.e., recommendation for using the CPG in practice) showed that all four high-quality CPGs were recommended by all reviewers [35,36,37,38]. All included CPGs had cited SRs in their reference list. The SIGN CPG (n = 25) had the most SR citations, including 12 Cochrane SRs, followed by NICE (n = 13), which also had one Cochrane SR, NZ (n = 6), which had three Cochrane SRs, and ACRC (n = 5), which had none [35,36,37,38].

### 3.3. Inter-Rater Analysis

The Appraisal of Guidelines for REsearch and Evaluation (AGREE II) critical group appraisal in Table 2 shows the critical group evaluation of the four selected recommendations against each domain by Research and Evaluation (AGREE II). We calculated the percentage of agreement between raters. The results of the inter-rater reliability tests showed a high degree of agreement between the four raters for each item in each of the six domains, as well as the percent agreement of the overall assessment 1 (OA1). The calculated AGREE II domain scores are displayed in Figure 1 and Figure 2.

Figure 2 shows that the majority of the Kappa scores ranged from 0.50 to 1.00, indicating good to excellent agreement. Four assessments, presented only in Figure 2, found a low level of agreement (K = 0.0): D3Q9 of SIGN 2016, D2Q5, D3Q10 of New Zealand (NZ MOH 2016), and D3Q8 of Autism CRC 2018. The SIGN 2016 review revealed 4 questions with excellent agreement (K = 1), D1Q1, D3Q7, D3Q13, and D3Q14; 7 with very good agreement (K = 0.75), D1Q2, D1Q3, D3Q12, D4Q15, D5Q19, D5Q20 and D6Q23; 11 with good agreement (K = 0.5), D2Q4, D2Q5, D2Q6, D3Q8, D3Q10, D3Q11, D4Q16, D4Q17, D5Q18, D5Q21 and D6Q22; 1 with poor agreement (K = 0.00), D3Q9; and the overall rating (1) with good agreement (K = 0.5).

The NICE 2017 evaluation revealed 4 questions with excellent agreement out of 24, D1Q3, D3Q14, D4Q15 and D5Q19; 14 with very good agreement (K = 0.75), D1Q1, D1Q2, D2Q4, D2Q6, D3Q7, D3Q11, D3Q12, D3Q13, D4Q16, D4Q17, D5Q20, D5Q21, D6Q22, and D6Q23; 5 with good agreement (K = 0.5), D2Q5, D3Q8, D3Q9, D3Q10, and D5Q18; and the overall rating (1) with very good agreement (K = 0.75). However, the New Zealand evaluation (NZ MOH 2016) revealed only 2 questions with poor agreement; 1 question with excellent agreement (K = 1) out of 24 questions, D5Q19; no questions with fair agreement; 9 questions with good agreement (K = 0.5), D1Q2, D2Q4, D2Q6, D3Q7, D3Q11, D5Q18, D5Q21, D6Q22, and D6Q23; eleven questions with very good agreement (K = 0.75), D1Q1, D1Q3, D3Q8, D3Q9, D3Q12, D3Q13, D3Q14, D4Q15, D4Q16, D4Q17, and D5Q20; and the overall assessment (1) with very good agreement (K = 0.75).

Finally, the Australian guideline evaluation (Autism CRC 2018) revealed that 1 question out of 24 had poor agreement; nine had good agreement (K = 0.5), D3Q9, D3Q11, D3Q13, D3Q14, D4Q15, D4Q16, D5Q18, D5Q20, and D5Q21; eight had very good agreement (K = 0.75), D1Q3, D2Q5, D3Q7, D3Q10, D3Q12, D4Q17, D6Q22, and D6Q23; and the overall assessment (1) had good agreement (K = 0.5). Appendix A shows the intraclass correlation coefficient (Kappa value) among raters for the four recommendations for overall assessment 2. The number of observed agreements was six (47.26 percent of the observations), and eight agreements were predicted by chance (65.00 percent of the observations) (Kappa = 0.637; kappa SE = 0.732; 95% confidence interval: Weighted Kappa = 0.082 for values ranging from −0.112 to 0.348).

## 4. Discussion

### 4.1. Early Identification and Diagnosis

The present study demonstrated the high quality of ASD guidelines and, at the same time, differences in terms of specificity of recommendations. All guidelines were explicit on the importance of early identification; however, different approaches were proposed. For example, NICE recommends that a local autism strategy group develop a local autism pathway for recognition and referral. On the other hand, the three other guidelines emphasized early identification of children with ASD, focusing on the basic tools and screening measures. NICE and ACRC were the only guidelines recommending using a multidisciplinary team representative of all healthcare professionals. Similarly, two CPGs advocated for delivering diagnostic assessment information to families (ACRC, SIGN). They made specific suggestions, such as providing information about the assessors, the names of the assessment tools (ACRC), and descriptions of what the tools measure (ACRC).

Three CPGs advised addressing general information on ASD with families (NICE, SIGN, NZ) and/or offering resources for families to acquire this information (NICE, SIGN, NZ). With the family’s permission, half of the CPGs recommended providing copies of the ASD medical report to the family and others, such as the referral clinician (ACRC), the child’s general practitioner or pediatrician (NICE), and other relevant professions or local health care providers (NICE, ACRC). When appropriate, the ACRC also recommended distributing the report to funding or service organizations.

### 4.2. Interventions and Management of ASD

Interestingly, there was limited information in the Australian guidelines regarding the intervention section of the guidelines. However, the NZ evaluation emphasized more on local issues that could be not applicable in other countries (although they can be applied to minorities anywhere in the world). The NZ evaluation addresses behavioral management and offers essential recommendations for treating and managing ASD. Only NICE and NZ mentioned the non-pharmacological intervention in detail, which is missing in the other guidelines. NICE had more details that serve many aspects of the intervention part of the guidelines.

Furthermore, the Australian evaluation does not include pharmacological intervention in the guidelines. However, NICE has more specific and detailed information regarding the use of medication, where SIGN and NZ mentioned their recommendations in a brief and general statement manner.

In terms of medication use, in the NICE and NZ guidelines, the only medication in the pharmacological part that is mentioned is the antipsychotic, and there was a clear statement of what not to use regarding the other medication. Antipsychotic use was mentioned in all guidelines (except the Australian one, which did not mention medication). There was a clear emphasis on sleep management in NICE with a brief description in SIGN and NZ. Additionally, SIGN was the only guideline that emphasized the psych-education for the family.

### 4.3. The transition of Care and Community Support

For those looking for guidelines related to transitions, NICE has more details guiding the care in this important step in the lives of young people with autism. SIGN elaborates on the nonpharmacological interventions, including parent-mediated interventions, communication intervention, social communication and interaction intervention, and behavioral interventions. NZ has exclusive details related to living in the community, leisure and recreation, and care when in contact with the justice system. There was very brief mention related to sensory integration intervention and occupational therapy in SIGN only. In comparison, only the Australian guidelines mentioned the professional discipline of occupational therapy.

### 4.4. Challenging Behaviors

There is a clear gap in crisis management guidelines and support with challenging behaviors in the majority of the guidelines. However, NZ guidelines have only a brief summary of recommendations related to crisis management and mental health and forensic services. Interestingly, nothing was mentioned related to school placement and interventions in any of the guidelines. The mention of the medications is minimal and needs more elaboration and details to guide the end-user in medication use in a straightforward matter. There was no mention of the side effects and counseling the parents. There is little mention of recommended programs for intervention, such as PECS.

### 4.5. At Local and International Levels

Despite the publication of several reviews of ASD CPGs [25,39,40], there are no evidence-based CPGs for ASD in Saudi Arabia; therefore, doctors are forced to make the best decisions for ASD patients based only on their clinical knowledge and experience. This review adds to the knowledge base using the AGREE II instrument to systematically assess the quality of recently published CPGs of ASD in children as part of a national CPG adaptation initiative in Saudi Arabia.

Two similar SRs of ASD CPGs were published during the undertaking of our research for the CPG project [39,40].

Using the AGREE II instrument, Wickstrom et al. evaluated the quality of 36 CPGs for ASD and 47 CPGs for intellectual disability published up until January 2019. However, only scores of individual items were reported rather than the domain standardized scores. The authors did not report the two overall assessments. Moreover, they mapped the biomedical, pharmacologic, and psychosocial interventions within the assessed CPGs [39]. Similarly, Pattison et al. assessed 17 ASD CPGs published up to April 2020. Because most of the recommendations were expert-based, with limited empirical research in this area, the authors observed variations in the recommendations for providing results and diagnoses to families and caregivers for children with ASD. They have included the four CPGs identified in our study and given them similar ratings in the six AGREE II domains, an additional strength of these four high-quality CPGs (ACRC, NICE, NZ, SIGN) [40].

### 4.6. Strengths and Limitations

Several strengths can be identified in our study. The authorship group of this SR and quality assessment included the aforementioned multidisciplinary clinical team with expertise in providing the different aspects of care for children with ASD in addition to a CPG methodologist. The results of this SR are enhanced by using a validated CPG appraisal instrument, the AGREE II. Moreover, the findings of our SR can inform groups who develop or adapt ASD CPGs.

There are numerous limitations to our study as well. First, only CPGs published in English or Arabic were included. High-quality CPGs and recommendations published in other languages are likely to have been overlooked in our review.

Furthermore, we did not utilize the new ‘AGREE-REX’ (Recommendation Excellence) tool, which addresses some of the shortcomings of the original AGREE II by addressing the clinical credibility of CPG recommendations [41]. Another potential limitation is the use of 70% as a cutoff point for standard domain ratings, although the original AGREE II does not mandate such a cutoff. Several cutoffs ranging from 50 to 80% have been used, but there is insufficient evidence to infer that one cutoff is superior to the other [42]. Finally, all of the four CPGs were from high-income countries.

## 5. Conclusions

All guidelines were of high quality; however, they differed in terms of the details of recommendations which could be attributed to the differences in healthcare settings. The findings of our study suggest that clinicians’ capacity building should include how to use the AGREE II instrument or at least the short AGREE Global Rating Scale (AGREE-GRS) instrument to support their decisions when selecting CPGs for use in everyday practice.

## Figures and Tables

**Figure 1 children-09-01050-f001:**
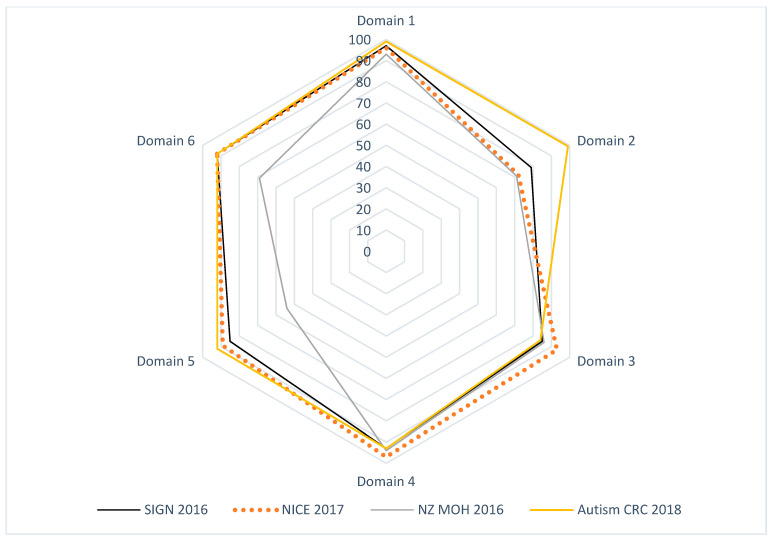
Radar map of the AGREE II final standardized domain scores for eligible appraised ASD CPGs.

**Figure 2 children-09-01050-f002:**
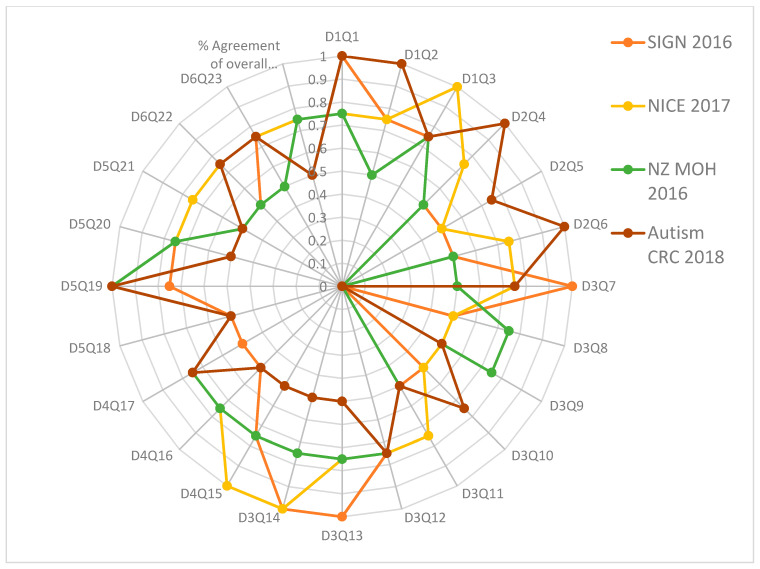
Percent agreement among raters for the four ASD clinical practice guidelines focusing on every question or item in every AGREE II domain.

**Table 1 children-09-01050-t001:** Characteristics of included ASD CPGs.

Organization, Country (Abbreviation)	Health System, Economic Classification	CPG Title	Year of Publication
Australian Autism CRC, Australia (ACRC)	National Health Insurance, High-income country	Australian National Guideline for the Assessment and Diagnosis of Autism Spectrum Disorders	2018
Ministry of Health New Zealand, New Zealand (NZ)	National Health Insurance, High-income country	New Zealand Autism Spectrum Disorder Guideline	2016
National Institute for Health and Care Excellence, United Kingdom (NICE)	National Health Service, High-income country	Autism spectrum disorder in under 19s: recognition, referral and diagnosis	2017
Scottish Intercollegiate Guidelines Network (SIGN), Healthcare Improvement Scotland, Scotland (HIS)	National Health Service, High-income country	Assessment, diagnosis and interventions for autism spectrum disorders	2019

**Table 2 children-09-01050-t002:** AGREE II assessment results and domain scores for the four included CPGs *.

Source CPGs/AGREE II Domains Scores (%)	ACRC 2018 [35]	NZ 2016 [36]	NICE 2017 [37]	SIGN 2019 [38]
**Domain 1. Scope and Purpose** Items 1–3: Objectives; health question(s); population (patients, public, etc.)	99%	93%	96%	97%
**Domain 2. Stakeholder Involvement** Items 4–6: Group membership; target population preferences and views; target users	99%	71%	72%	79%
**Domain 3. Rigor of development** Items 7–14: Search methods; evidence selection criteria; strengths and limitations of the evidence; formulation of recommendations; consideration of benefits and harms; link between recommendations and evidence; external review; updating procedure.	84%	86%	93%	85%
**Domain 4. Clarity and presentation** Items 15–17: Specific and unambiguous recommendations; management options; identifiable key recommendations	93%	94%	97%	93%
**Domain 5. Applicability** Items 18–21: Facilitators and barriers to application; implementation advice/tools; resource implications; monitoring/auditing criteria	92%	54%	89%	85%
**Domain 6. Editorial independence** Items 22, 23: Funding body; competing interests	92%	69%	92%	92%
**Overall Assessment 1** (Overall quality)	92%	88%	96%	92%
**Overall Assessment 2** (Recommend the CPG for use by four appraisers)	Yes—3, Yes with modifications—1, No—0	Yes—4, Yes with modifications—0, No—0	Yes—2, Yes with modifications—2, No—0	Yes—2, Yes with modifications—2, No—0

* AGREE II (Appraisal of Guidelines for REsearch and Evaluation Version II Instrument); Australian Autism CRC (ACRC); Ministry of Health New Zealand (NZ); National Institute for Health and Care Excellence (NICE); Scottish Intercollegiate Guidelines Network, Healthcare Improvement Scotland (SIGN-HIS). This AGREE II domain scores table is a standard template with a constant set of attributes that have been reported in most similar reports of AGREE II assessment of guidelines (example: Alhasan, K.A., Al Khalifah, R., Aloufi, M. et al. AGREEing on clinical practice guidelines for idiopathic steroid-sensitive nephrotic syndrome in children. Syst Rev 10, 144 (2021). https://doi.org/10.1186/s13643-021-01666-w).

## Data Availability

The data that support the findings of this study have been made available in the tables, figures, and appendices of this article. Further details could be made available from the authors upon reasonable request to the corresponding author.

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
