# Peer review of "AGREEing on Clinical Practice Guidelines for Autism Spectrum Disorders in Children: A Systematic Review and Quality Assessment"

_children, 2022, doi:10.3390/children9071050_

Round 1
Reviewer 1 Report
The article contains interesting research results about using AGREE II instrument in the evaluation of the quality of clinical practice guidelines for ASD children. This study could be helpful for researchers and practitioners in assessing and using AGREE II.
The introduction section is well written and concise.
The material and methods section is appropriate and informative.
Results were well-presented and instructive.
Discussion.
It would be noteworthy to know how differences in various health care settings are reflected in ASD CPGs.
Author Response
Thank you for your valuable feedback.
Reviewer 2 Report
Manuscript title: AGREEing on clinical practice guidelines for autism spectrum disorders in children: a systematic review and quality assessment
My comments:
General comment: Thank you for the opportunity to review this manuscript draft. The authors performed a systematic review on the clinical practice guidelines (CPGs) for autism spectrum disorder (ASD) in children. This is a great effort involving a team of authors. However, there a few areas of concern to be addressed before it could be published.
Title: The title is interesting and catchy, so it should be able to attract readers.
Abstract: As a background, it was mentioned that “Autism Spectrum Disorder (ASD) is a neurodevelopmental disorder with symptoms appearing as early as 18 months of age. Children with ASD frequently have psychological and/or somatic co-morbid conditions”.
I have 2 comments: 1) The background should highlight the needs for this review, rather than providing the general information on ASD. 2) To say that children on the spectrum frequently have somatic co-morbid conditions is inaccurate.
Introduction:
1) Page 2 (Line 70-74). The reference you mentioned here (ref number 2) does not seem to contain the findings in the in-text citations. Please check
2) Page 2 (Line 97-99): This is confusing – “of this” referred to children with ASD, or children in the cited study?
3) Page 3 (Line 108): Please check this in-text citation format: Becerra-Culqui et al., 2018 and Hill et al., 2021,
4) Page 3 (Line 114) : The first use of the acronym “CPGs” was not define here.
5) Overall, there are several grammatical errors, particularly on the use of articles, for example: Page 3 (Line 128):
We have dedicated this study to reporting the outcomes of this SR and critically evaluating the recently released CPGs for children with ASD using Appraisal of Guidelines for REsearch and Evaluation (AGREE) II as it is a core step in the CPG adaptation process.
Another example is in Page 4 (Line 163): ..abstract of the retrieved.., and Page 11 (Line 459) “the details of recommendations”
Please check through the manuscript for these errors.
Methods:
1) Page 4 (Line 159-160): What was the rationale behind the time range chosen for this study?
2) The PIPOH and PICAR acronyms should be written in full. Furthermore, a brief description on these models would help reader understand the review better.
3) Page 4 (Line 172): The eligibility criteria of “accessible from a CPG database or peer-reviewed journal” – is it separate from eligibility criterion number 5? If so, making it the sixth criterion looks better.
4) Page 5 (Line 225) typo “ICCC”.
Results: Page 5 (Line 238): 161 being excluded based on the health questions and eligibility criteria. What do the authors mean by health questions?
Discussion: This was done briefly and straight-to-the-point. I have no comment.
Strength and Limitations:
1) Page 11 (Line 447-448): “Furthermore, the new ‘AGREE-REX’ (Recommendation EXcellence) tool addresses some of the shortcomings of the original AGREE II by addressing the clinical credibility of CPG recommendations” – how is this a shortcoming?
2) Selection of the final four CPGs from high-income countries should be discussed as a limitation.
Conclusion: The first and second sentence appears to be not in order (the second should be the first).
References: Please check thoroughly and ensure uniformity in the format of referencing.
Author Response
Main points to address:
Abstract:
As a background, it was mentioned that “Autism Spectrum Disorder (ASD) is a neurodevelopmental disorder with symptoms appearing as early as 18 months of age. Children with ASD frequently have psychological and/or somatic co-morbid conditions”.
I have 2 comments: 1) The background should highlight the needs for this review, rather than providing the general information on ASD. 2) To say that children on the spectrum frequently have somatic co-morbid conditions is inaccurate.
Response:
Thanks for the comment. We agree and the abstract was updated accordingly.
Introduction:
1) Page 2 (Line 70-74). The reference you mentioned here (ref number 2) does not seem to contain the findings in the in-text citations. Please check
Response:
Thanks for your comment. We agree and the manuscript was updated accordingly.
2) Page 2 (Line 97-99): This is confusing – “of this” referred to children with ASD, or children in the cited study?
Response:
Thanks for your comment. We agree and the manuscript was updated accordingly.
3) Page 3 (Line 108): Please check this in-text citation format: Becerra-Culqui et al., 2018 and Hill et al., 2021,
Response:
Thanks for your comment. We agree and the manuscript was updated accordingly.
4) Page 3 (Line 114) : The first use of the acronym “CPGs” was not define here.
Response:
Thanks for your comment. We agree and the manuscript was updated accordingly.
5) Overall, there are several grammatical errors, particularly on the use of articles, for example: Page 3 (Line 128):
We have dedicated this study to reporting the outcomes of this SR and critically evaluating the recently released CPGs for children with ASD using Appraisal of Guidelines for REsearch and Evaluation (AGREE) II as it is a core step in the CPG adaptation process.
Response:
Thanks for your comment. We agree and the manuscript was updated accordingly.
Another example is in Page 4 (Line 163): ..abstract of the retrieved.., and Page 11 (Line 459) “thedetails of recommendations”
Please check through the manuscript for these errors.
Response:
Thanks for your comment. We agree and the manuscript was updated accordingly.
Methods:
1) Page 4 (Line 159-160): What was the rationale behind the time range chosen for this study?
Response:
Thanks for your comment. We have decided the period of time to update the search conducted in the previous SR of CPGs as the last systematic review was conducted in 2018. We agree and the manuscript was updated accordingly.
2) The PIPOH and PICAR acronyms should be written in full. Furthermore, a brief description on these models would help reader understand the review better.
Response:
Thanks for your comment. We agree and the manuscript was updated accordingly.
3) Page 4 (Line 172): The eligibility criteria of “accessible from a CPG database or peer-reviewed journal” – is it separate from eligibility criterion number 5? If so, making it the sixth criterion looks better.
Response:
Thanks for your comment. We agree and the manuscript was updated accordingly.
4) Page 5 (Line 225) typo “ICCC”.
Response:
Thanks for your comment. We agree and the manuscript was updated accordingly.
Results: Page 5 (Line 238): 161 being excluded based on the health questions and eligibility criteria. What do the authors mean by health questions?
Response:
Thanks for your comment. We mean based on PIPOH health questions. The manuscript was updated accordingly to clarify its meaning.
Discussion: This was done briefly and straight-to-the-point. I have no comment.
Response:
Thanks for your comment.
Strength and Limitations:
1) Page 11 (Line 447-448): “Furthermore, the new ‘AGREE-REX’ (Recommendation EXcellence) tool addresses some of the shortcomings of the original AGREE II by addressing the clinical credibility of CPG recommendations” – how is this a shortcoming?
Response:
Thanks for your comment. We just wanted to highlight that we did not utilize the new AGREE-REX tool which is more extensive but needs more resources.
2) Selection of the final four CPGs from high-income countries should be discussed as a limitation.
Response:
Thanks for the comment. We agree and the manuscript was updated accordingly.
Conclusion: The first and second sentence appears to be not in order (the second should be the first).
Response:
Thanks for the comment. We agree and the manuscript was updated accordingly.
References: Please check thoroughly and ensure uniformity in the format of referencing.
Response:
Thanks for the comment. We agree and the reference list was updated accordingly.